# The Socioeconomic Inequality in Increment of Caries and Growth among Chinese Children

**DOI:** 10.3390/ijerph17124234

**Published:** 2020-06-13

**Authors:** Anqi Shen, Eduardo Bernabé, Wael Sabbah

**Affiliations:** 1Department of Preventive Dentistry, Beijing Stomatology Hospital, Capital Medical University, 4th Tiantanxili, Dongcheng District, Beijing 100050, China; 2Faculty of Dentistry, Oral & Craniofacial Sciences, King’s College London, London SE5 9RS, UK; eduardo.bernabe@kcl.ac.uk (E.B.); wael.sabbah@kcl.ac.uk (W.S.)

**Keywords:** dental caries, growth, socioeconomic factors, child, longitudinal studies

## Abstract

Background: This study aimed at assessing socioeconomic inequalities in the increment of dental caries and growth among preschool Chinese children, and to assess the best predictor of socioeconomic inequality for each of these conditions. Methods: This is a longitudinal population-based study. The sample data included preschool children living in three cities of the Liaoning Province, China. At baseline, 15 kindergartens with 1111 participants were included and dropped to 772 with a response rate of 70% at follow-up. Mean ages at baseline and follow-up were 50.82 and 60.55 months, respectively. Median follow-up time was 10.12 months. Data were collected through structured questionnaire, oral examination and anthropometric measurement. The questionnaire collected information on sex, age, family income, mother’s education and children’s dietary habits. The numbers of decayed, missing and filled teeth (DMFT) was used to indicate dental caries. Weight- and height-for-age z-scores were calculated using the WHO Growth Standard. Multilevel analysis was used to assess the association between baseline socioeconomic position and each of dental caries and child’s growth. Results: Mother’s education was negatively associated with increments of DMFT. Family income was not significantly associated with DMFT in the fully adjusted model. The association persisted after accounting for other socioeconomic and dietary factors. Higher income was positively related to an increase in the weight-for-age z-score. The relationship between income and changes in the height-for-age z-score was positive and significant in the second highest income group. Conclusions: Mother’s education appeared to be the strongest predictor of increments of dental caries. Only income was significantly associated with an increase in children’s weight and height.

## 1. Introduction

Dental caries is a diet-related condition and is one of the most common oral diseases in China, with almost half of the children and majority of the adults affected [1]. Despite the decline of dental caries in Western countries in the past decades, inequalities in oral health still exist in most of the countries [2]. The high-level of caries in China is mainly attributed to the increased consumption of sugars, lack of dental services, and socioeconomic factors, particularly income and parental education [3].

Several studies found significant associations between lower socioeconomic position and higher risk of dental cavities [4]. In China, there were significant inequalities in the sum of decayed, missing and filled teeth (DMFT) index by geographical location among 12-year-old children [3] and in DMFT index by income among 5-year-old children [5]. Furthermore, parental education was found to be inversely related to children’s caries experiences [6]. Parental education and income appear to affect children’s oral health through an impact on knowledge of oral health risk behaviors and attitudes towards prevention, such as purchasing dental hygiene products, toothbrushing habits, frequency and patterns of health service utilization, all linked to children’s oral health [7]. Furthermore, parental level of education allows a better understanding of information that enables them to care for their children which may affect their children’s oral health [8]. Higher mother education and income may also affect children’s dietary habits, which could lead to a positive impact on dental caries. 

On the other hand, the inequality in socioeconomic position might affect the dietary habits and nutritional intake of Chinese children, which could also affect children’s growth (weight and height). With the increase in income inequalities in China over the past 25 years, it was argued that health inequalities in China are attributed to demographic and socioeconomic factors, among which income, occupation and education are the most important ones [9]. Yao’s (2019) study indicated that there is an association between income inequality and BMI index [10]. Other studies also showed strong a socioeconomic gradient in stunting [11] and thinness [12]. Children from families with lower income are more likely to buy low-cost food and have a poor dietary intake, thus decreasing essential nutrients intake such as protein and fat, which increase the risk of being underweight and stunting [13]. These apparent income inequalities indicate that a reduction in economic inequalities could reduce inequalities in undernutrition [14]. 

Given the common role of socioeconomic factors in both dental caries and undernutrition, this longitudinal study set out to examine whether the same socioeconomic indicators are linked to changes in dental caries and children’s growth in the same population. The objectives of this study are to examine socioeconomic inequalities in increment of dental caries and growth among preschool Chinese children, and to assess the strongest predictor of socioeconomic inequality for each of these conditions.

## 2. Materials and Methods 

Ethical approval was obtained for King’s College London (KCL Ethics Ref: HR-15/16–2901), and Shenyang Dental Hospital gave the oral consent (Ministry of Health of People’s Republic of China). Children whose parents agreed to participant were included at baseline and follow-up in this study. The sample size calculation of 636 is based on a previous study among preschool children in Hong Kong [15]. The number was increased to 1000 to account for dropouts. 

### 2.1. Study Population

This is a longitudinal population-based study, and data was collected at two points in time. The study recruited preschool children in the cities of Shenyang, Liaoyang and Fushun, Liaoning Province, China. At baseline, 15 kindergartens (eight kindergartens in rural area and seven kindergartens in urban area) with 1111 participants were included in this study. At follow-up, the number of participants was 772 with a response rate of 70% at follow-up. The participants who were lost at follow-up either left the kindergarten, did not complete clinical examination or did not return a signed consent form. Mean ages of participants at baseline and follow-up were 50.82 and 60.55 months, respectively. Median and mean follow-up times were 10.12 and 9.73 (SD: 1.22) months, respectively.

### 2.2. Data Collection

Dental examination and anthropometric measurement were assessed at baseline and follow-up. A structured questionnaire was used to collect information on demographic factors, family income, mother’s education, and children’s dietary habits. The questionnaire was based on the WHO questionnaire [16] and modified following a Chinese questionnaire used in national oral health surveys. In this analysis, baseline data from the questionnaire were in the analysis. 

Sex was coded as male and female, and area was coded as urban and rural area. Income was collected using five options: 0–3999 RMB, 4000–5999 RMB, 6000–9999 RMB, 10,000 RMB or above and undeclared. Income groups were based on the 4th Chinese Oral Health Survey [17]. Mother’s education was grouped as primary and middle school, high school and junior college, bachelor or above, and others. The parents reported the intake frequency of five food or drinks of their children: biscuit/cake/bread, candy/chocolate, fizzy drinks and fruit juice on 6-point response scales (twice or above per day, once per day, 3–4 times per week, once per week, once per month, or seldom or never). Weighted scores were calculated to reflect the frequency of consumption in each response category. Weighted scores of four snacks and drinks were then summed up to produce an overall score [3] which was used to reflect daily frequency of sugar intake. Frequency of fresh fruit consumption was calculated in the same manner and was used as a separate variable. 

The survey team of oral examination included two persons, a dentist and a recorder. The dentist was trained on data collection using WHO standard criteria for assessing dental caries [16]. Eighty participants were re-examined in different days to calculate intra-examiner reliability. Agreement level (Kappa) was 0.72, indicating substantial agreement. Children with an urgent need for treatment were referred to a dentist. The sum of decayed, missing and filled teeth (DMFT) was used to examine the dental caries level of participants. 

One health worker was responsible for assessing weight and height and was blinded to the dental condition of the participants. Height of the child was assessed with a weight and height scale to the nearest 0.5 cm. Similarly, weights were assessed to the nearest 0.5 kg. Children wore light clothes and removed shoes before they stood on the scale. Age- and sex-specific percentile for weight and height was used to indicate body measurements of the children [18]. The WHO Growth Standard’s for 2006 and 2007 were also used [19,20]. The WHO Growth Standards 2006 was utilized to convert weight and height measurements to weight-for-age and height-for-age z-scores among children aged 0–5 years old and WHO Growth Standards 2007 was used for children older than 5 years [19,20]. 

### 2.3. Statistical Analysis

STATA (Stata Statistical Software: Release 15. TX: StataCorp LLC, College Station, TX, USA) was used to analyze the data. Multilevel analysis was used to assess the association between baseline socioeconomic position and dental caries, weight and height. Participants who had two clinical assessments were included in the analysis. Firstly, the distributions of baseline demographic (age, and sex), socioeconomic factors (income and mother’s education), area (urban/rural) and dietary habits (mean fresh fruit and mean sugar consumption) were assessed. Then distributions of DMFT index, weight-for-age z-score (WAZ) and height-for-age z-score (HAZ) at baseline and follow-up were assessed. 

Secondly, multilevel analysis was used to explore the association between baseline socioeconomic position and changes in dental caries. Three models were constructed to assess the association between changes in DMFT and each of income, education and area, accounting for time and sex. The fully adjusted model included all socioeconomic indicators, demographic factors and diet. Similar sets of multilevel models were constructed for each of change in height and weight. 

## 3. Results

Mean age of those lost at follow-up was 51.15 months at baseline, 51.10% were boys, mean DMFT was 3.17, and means of actual weight and height were 18.56 and 106.26, respectively. In these parameters, differences between those who were lost at follow-up and those included in the analysis were minimal. Details of the study is published elsewhere [21]. 

Table 1 shows mean age at baseline (50.82) and follow-up (60.55) months. The percentage of boys was 51.42%, and 65.54% of children lived in the rural area. Only 14% of the participants were in the highest income group. The highest percentage of mother’s education was high school and junior college (30.70%). The mean DMFT index was 3.18 at baseline and 4.21 at follow-up. Mean weight z-score (WAZ) was 0.58 at baseline and 0.66 at follow-up. Mean height z-score (HAZ) was 0.49 at baseline and 0.69 at follow-up. 

Table 2 shows results from the multilevel analysis for changes in DMFT index. Area (rural) was positively and significantly associated with an increase in DMFT (coefficient: 0.86; 95% CI (confidence interval): 0.36, 1.37) in the model adjusting for sex and age. Lower mother education was significantly associated with an increase in DMFT score in the semi-adjusted and fully adjusted models with coefficient 0.96; 95% CI: 0.18, 1.74 and 0.86; 95% CI: 0.20, 1.51 for the lowest and second lowest education groups. Only the highest income group (10,000 RMB or above) was negatively and significantly associated with changes in DMFT index (−0.90, 95% CI: −1.77, −0.02) in the semi-adjusted model. However, the association was insignificant in the fully adjusted model.

Table 3 shows results from the multilevel analysis for changes in weight-for-age z-score (WAZ). In the semi-adjusted model, higher income was associated with an increase in weight-for-age z-score. After adjusting for all socioeconomic and dietary factors, only the second lowest income group was significantly associated with weight (coefficient: 0.36; 95% CI: 0.11, 0.61). Table 4 shows the association between changes in height-for-age z-score (HAZ) and socioeconomic factors (income, mother’s education and area). The relationship between income and changes in height-for-age z-score was positive and significant in the second highest income group (coefficient: 0.26; 95% CI: 0.03, 0.49) in the semi-adjusted model. 

## 4. Discussion

In this study we assessed socioeconomic inequalities in both of children’s growth and dental caries in the same population of Chinese preschool children. The main findings showed that there was negative and significant association between income, mother’s education and changes in dental caries. Income, rather than mother’s education, was positively and significantly associated with weight and height. Children from rural areas had a greater increase in dental caries, a smaller increase in weight and height compared to those from urban areas.

The findings of the current analysis of dental caries inequalities among Chinese children are consistent with studies from other countries. Dental caries is related to lower socioeconomic conditions, such as parental education, family income and area [22]. Untreated decayed teeth increase consistently as the family income decreases [23], which is consistent with the finding of this study. The places where people live (urban or rural areas) have also been reported as a factor that influence the association between health outcomes and related factors [24]. Education was considered as the most important indicator of socioeconomic inequalities in children’s caries [25], a finding consistent with this study. In this study, mother’s education appeared to be a stronger predictor of children’s dental caries than family income or area. This is probably because highly educated mothers are more likely to have better oral health knowledge and to provide better care for the children, particularly related to behaviors such as diet, oral hygiene and use of dental services. Furthermore, mother’s education is also linked to income, which impacts on better living conditions and facilitates access to oral hygiene products, better diet and preventive services [26].

Several studies demonstrated that the level of caries is associated with deprivation [27]. Mothers in lower socioeconomic positions are more likely to allow their children to consume unhealthy food such as sugary, fatty and acidic food [28,29] and to select food that optimizes quantity rather than nutritional quality, compared to mothers in higher socioeconomic positions [30,31], which may result in dental caries. Moreover, children who belong to families at the bottom of the social hierarchy are more likely to have inadequate access to food, food shortages and inadequate eating patterns. These dietary patterns are likely to lead to an increased consumption of fermentable carbohydrate, a risk factor for dental caries [30,31]. These dietary habits are implicated in the mechanism linking socioeconomic factors and dental caries [30]. On the other hand, children living in countries and communities with higher socioeconomic conditions are more likely to consume food with high calories, snacks, soft drinks and even fermentable carbohydrates [32,33]. This could also contribute to an increased risk of dental caries and weight gain in developed countries [34]. 

Socioeconomic inequalities in dental caries are not only attributed to different patterns of food and drinks consumption, but also to other oral health behaviors such as toothbrushing habits [35], limited access to dental healthcare services, poor oral hygiene and poor living conditions, which could accelerate the development of decayed teeth [28]. 

Socioeconomic factors and material abilities are also strongly linked children’s growth [36]. Poorer families are more likely to experience poor dietary intake due to reduced access to healthy food containing less sugar and fat, and rich in essential nutrients such as milk and protein [13]. Poor diet also impacts children’s abilities to concentrate, sleep and attend schools, which could also contribute to undernutrition [28]. Low income is also associated with poor sanitation and poor hygiene that lead to increased infections in children, which detrimentally affect children’s development [37]. In the current study, there was consistent income inequality in weight gain than in height gain. This was consistent with findings by Wagstaff and Watanabe [38] who concluded that inequalities in being underweight tend to be larger than inequalities in stunting [38]. 

Unlike in developed countries, in China, obesity and overweight are more common among those with a higher socioeconomic status. Factors such as overconsumption of food among children in higher socioeconomic families in developing countries leads to the risk of excessive weight gain [36]. Economic wealth and income inequalities also impact childhood development as they usually use private transportation rather than walking and engage in sedentary entertainment such as video games, which influence their energy balance [39]. 

Sugar consumption is a leading risk factor for both caries and being overweight due to a higher energy intake [40]. However, in this study, sugar consumption was significantly associated with dental caries, but not with children’s growth. The lack of significant association between sugar consumption and anthropometric measures observed in this study could be because sugar is not the sole determinant of children’s growth. The nutritional transition in China has been linked to an increase in consumption of animal proteins [41]. This increase of energy intake is considered as a key determinant of children’s growth [42]. 

The strength of this study is in using longitudinal data to assess socioeconomic inequalities in both dental caries and children’s growth in the same population of preschool children in China. There are some limitations worth mentioning. Approximately 30% of the participants were lost at follow-up, mainly because the children were recruited from kindergartens and they could have moved from kindergartens as their parents changed jobs. Furthermore, those who left the study were to a great extent similar to those included in the analysis in terms of clinical outcomes and sociodemographic characteristics. Secondly, a longer time interval between the two assessments could have affected the findings; this however was not possible given the short period children spend in the kindergarten. Thirdly, only children who attended selected kindergartens were included in this study. These children are usually from families with higher income who can afford kindergarten fees. It is possible that if the study was conducted among the general population, greater inequalities could have been observed. 

## 5. Conclusions

The study demonstrated socioeconomic inequalities in changes in both dental caries and growth among preschool Chinese children. While mother’s education appeared to be the strongest predictor of dental caries, income was the only factor significantly associated with weight and height gain. 

## Figures and Tables

**Table 1 ijerph-17-04234-t001:** Description of all variables used in the analysis (*n* = 772).

Variables	N	Percentage/Mean	(95% CI)
Mean age at baseline	50.82 months		(50.09, 51.55)
Mean age at follow-up	60.55 months		(59.84, 61.26)
Gender			
Male	397	51.42%	(47.89, 54.95)
Female	375	48.58%	(45.05, 52.11)
Area			
Urban	266	34.46%	(31.18, 37.89)
Rural	506	65.54%	(62.11, 68.82)
Income ^a^			
0–3999	129	16.71%	(14.25, 19.52)
4000–5999	162	20.98%	(18.25, 24.01)
6000–9999	162	20.98%	(18.25, 24.01)
10,000 or above	108	13.99%	(11.71, 16.63)
Undeclared	211	27.34%	(24.30, 31.59)
Mother education ^a^			
Primary school and middle school	200	25.91%	(22.93, 29.12)
High school and junior college	237	30.70%	(27.54, 34.05)
Bachelor or above	216	27.98%	(24.92, 31.26)
Others	119	15.41%	(13.03, 18.14)
Mean Fresh fruit consumption ^a^		1.41	(1.37, 1.45)
Mean Sugar consumption ^a^		1.03	(0.97, 1.09)
Mean DMFT (baseline)		3.18	(2.91, 3.45)
Mean DMFT (follow up)		4.21	(3.90, 4.51)
Mean weight-for-age z-score (baseline)		0.58	(0.50, 0.66)
Mean weight-for-age z-score (follow up)		0.66	(0.58, 0.75)
Mean height-for-age z-score (baseline)		0.49	(0.42, 0.56)
Mean height-for-age z-score (follow up)		0.69	(0.62, 0.76)

^a^ Baseline variable.

**Table 2 ijerph-17-04234-t002:** Multilevel linear analysis of factors associated with changes in decayed, missing and filled teeth (DMFT) over one year among 772 preschool children in China.

	Model 1	Model 2	Model 3	Model 4
	Coefficient (95% CI)	Coefficient (95% CI)	Coefficient (95% CI)	Coefficient (95% CI)
Age (year)	1.29 *** (1.17, 1.41)	1.29 *** (1.17, 1.41)	1.29 *** (1.17, 1.41)	1.29 *** (1.16, 1.41)
Sex (Female)	0.44 (−0.03, 0.90)	0.41 (−0.05, 0.87)	0.39 (−0.07, 0.86)	0.42 (−0.04, 0.88)
Income				
0–3999 (reference)				
4000–5999	0.02 (−0.76, 0.80)			0.11 (−0.68, 0.89)
6000–9999	−0.57 (−1.34, 0.21)			−0.16 (−0.97, 0.65)
10,000 or above	−0.90 * (−1.77, −0.02)			−0.24 (−1.19, 0.70)
Undeclared	−0.35 (−1.10, 0.39)			−0.38 (−1.25, 0.50)
Mother education				
Bachelor or above (reference)				
Primary school and middle school		1.28 *** (0.62, 1.93)		0.96 * (0.18, 1.74)
High school and junior college		1.07 *** (0.48, 1.67)		0.86 * (0.20, 1.51)
Others		0.95 * (0.22, 1.69)		0.99 * (0.01, 1.98)
Area				
Urban (reference)				
Rural			0.86 ** (0.36, 1.37)	0.37 (−0.24, 0.98)
Fresh fruit	0.17 (−0.31, 0.65)	0.30 (−0.18, 0.78)	0.29 (−0.20, 0.77)	0.38 (−0.11, 0.88)
Mean sugar consumption	0.33 * (0.06, 0.59)	0.31 * (0.05, 0.58)	0.33 * (0.06, 0.59)	0.32 * (0.05, 0.58)

Model 1: adjusted for age, sex, income, fruit and sugar consumption. Model 2: adjusted for age, sex, mother education, fruit and sugar consumption. Model 3: adjusted for age, sex, area, fruit and sugar consumption. Model 4: adjusted for age, sex, income, mother’s education, area, fruit and sugar consumption. *** *p* < 0.001, ** *p* < 0.01, * *p* < 0.05.

**Table 3 ijerph-17-04234-t003:** Multilevel linear analysis of factors associated with changes in weight-for-age z-score (WAZ) over one year among 772 preschool children in China.

	Model 1	Model 2	Model 3	Model 4
	Coefficient (95% CI)	Coefficient (95% CI)	Coefficient (95% CI)	Coefficient (95% CI)
Age (year)	0.09 *** (0.05, 0.13)	0.09 *** (0.05, 0.13)	0.09 *** (0.05, 0.13)	0.09 *** (0.05, 0.13)
Sex (Female)	−0.22 ** (−0.37, −0.07)	−0.21 ** (−0.37, −0.06)	−0.22 ** (−0.37, −0.06)	−0.21 ** (−0.36, −0.06)
Income				
0–3999 (reference)				
4000–5999	0.35 ** (0.10, 0.60)			0.36 ** (0.11, 0.61)
6000–9999	0.26 * (0.01, 0.51)			0.26 (−0.01, 0.52)
10,000 or above	0.29 * (0.01, 0.57)			0.27 (−0.03, 0.58)
Undeclared	0.22 (−0.02, 0.46)			0.07 (−0.21, 0.36)
Mother education				
Bachelor or above (reference)				
Primary school and middle school		−0.06 (−0.28, 0.15)		0.03 (−0.23, 0.28)
High school and junior college		−0.06 (−0.26, 0.14)		−0.04 (−0.25, 0.18)
Others		0.08 (−0.16, 0.33)		
Area (reference)				
Urban				
Rural			−0.07 (−0.24, 0.09)	−0.04 (−0.24, 0.16)
Fresh fruit	−0.03 (−0.18, 0.12)	−0.02 (−0.18, 0.14)	−0.03 (−0.18, 0.13)	−0.03 (−0.19, 0.13)
Mean sugar consumption	−0.01 (−0.10, 0.07)	−0.14 (−0.10, 0.07)	−0.01 (−0.10, 0.07)	−0.01 (−0.10, 0.08)

Model 1: adjusted for age, sex, income, fruit and sugar consumption. Model 2: adjusted for age, sex, mother education, fruit and sugar consumption. Model 3: adjusted for age, sex, area, fruit and sugar consumption. Model 4: adjusted for age, sex, income, mother’s education, area, fruit and sugar consumption. *** *p* < 0.001, ** *p* < 0.01, * *p* < 0.05.

**Table 4 ijerph-17-04234-t004:** Multilevel linear analysis of factors associated with changes in height-for-age z-score (HAZ) over one year among 772 preschool children in China.

	Model 1	Model 2	Model 3	Model 4
	Coefficient (95% CI)	Coefficient (95% CI)	Coefficient (95% CI)	Coefficient (95% CI)
Age (year)	0.21 *** (0.18, 0.24)	0.21 *** (0.18, 0.24)	0.21 *** (0.18, 0.24)	0.21 *** (0.18, 0.24)
Sex (Female)	−0.13 (−0.27, 0.01)	−0.12 (−0.26, 0.02)	−0.12 (−0.26, 0.02)	−0.12 (−0.26, 0.02)
Income				
0–3999 (reference)				
4000–5999	0.19 (−0.04, 0.42)			0.18 (−0.05, 0.41)
6000–9999	0.26 * (0.03, 0.49)			0.21 (−0.03, 0.46)
10,000 or above	0.24 (−0.02, 0.49)			0.17 (−0.11, 0.45)
Undeclared	0.09 (−0.13, 0.31)			0.01 (−0.26, 0.26)
Mother education				
Bachelor or above (reference)				
Primary school and middle school		−0.18 (−0.38, 0.02)		−0.08 (−0.32, 0.15)
High school and junior college		−0.07 (−0.26, 0.11)		−0.03 (−0.23, 0.17)
Others		−0.08 (−0.30, 0.14)		0.09 (−0.20, 0.39)
Area				
Urban (reference)				
Rural			−0.13 (−0.28, 0.03)	−0.06 (−0.24, 0.13)
Fresh fruit	0.01 (−0.13, 0.15)	−0.01 (−0.14, 0.14)	−0.01 (−0.14, 0.14)	−0.01 (−0.16, 0.13)
Mean sugar consumption	0.01 (−0.08, 0.08)	0.01 (−0.08, 0.08)	0.01 (−0.08, 0.08)	0.01 (−0.08, 0.08)

Model 1: adjusted for age, sex, income, fruit and sugar consumption. Model 2: adjusted for age, sex, mother education, fruit and sugar consumption. Model 3: adjusted for age, sex, area, fruit and sugar consumption. Model 4: adjusted for age, sex, income, mother education’s, area, fruit and sugar consumption. *** *p* < 0.001, * *p* < 0.05.

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
