# Peer review of "The Socioeconomic Inequality in Increment of Caries and Growth among Chinese Children"

_ijerph, 2020, doi:10.3390/ijerph17124234_

Round 1

Reviewer 1 Report

Thank you for the opportunity to review this manuscript. I think that this is an important topic, that can provide useful information moving forward. 

After reading the manuscript, my major points for revision are mostly centered around methods for data collection and analysis. I think some re-tooling is needed for statistical approach, and that authors may benefit from restructuring this substantially. My other comments are below.

Was there a sample size calculation?

Line 90: What were income quintiles based on?

Line 92: Syntax/grammatical issues

Line 95: Were these weighted by quantity of sugar x frequency?

Line 94-96: Was this FFQ for a discrete time period, or generally? Recall bias is different depending on the way that the FFQ is administered. Was this a culturally appropriate FFQ? Was it validated, or how did foods end up on query?

Line 128: Please include a diagram outlining participants included and excluded, with drop out reasons and demographics for children that might skew sample conclusions

Table 2: This could benefit from reorganization for visual appeal

I don’t know that the socioeconomic/intake index against anthropometric data is that important to show. This (anthropometric outcomes) could be assessed for covariance with dental caries scores through regression analysis, couldn’t they? What is the further impact of this data?

Also, I do think that anthropometric outcomes need to be considered in modeling for the DMTF scores, as they may clearly have impact on baseline/ baseline to endline outcomes.

Can authors consider doing a time modeling adjustment for DMTF outcomes and covariates?

I think the analysis could use work overall: authors may benefit from looking at baseline outcomes, then endline outcomes adjusted for baseline, including anthropometrics, then change in baseline to endline outcomes (sort of like how researchers are looking at growth velocity now)

Author Response

Response to reviewer’s comments:

We would like to thank the reviewer for the extensive effort in reviewing this paper. We have now addressed all the new reviewer’s comments and prepared a point by point response. Please find our response to your comments.

  1. Was there a sample size calculation?

Response to reviewer: Yes. Sample size calculation is included in the methods. (Page 2, lines 70, 71) “Children whose parents agreed to participant were included at baseline and follow-up in this study. The sample size calculation is based on a previous study among preschool children in Hong Kong was 636. The number was increased to 1000 to account for dropouts.”

  1. Line 90: What were income quintiles based on?

Response to reviewer: We used income groups similar to those used in the Chinese national oral health survey (4th Oral Health Survey) “Lu, Hai Xia, et al. "The 4th National Oral Health Survey in the mainland of China: background and methodology." Chin J Dent Res 21.3 (2018): 161-5”.

  1. Line 92: Syntax/grammatical issues

Response to reviewer: We have corrected this.

  1. Line 95: Were these weighted by quantity of sugar x frequency?

Response to reviewer: This weighting accounts for the frequency of eating sugar per day. Data about fresh fruit and sugar intake frequency was extracted from the question “How often does your child eat or drink any of the following foods, even in small quantities?”. The parents reported the intake frequency of five food or drinks: fresh fruit, biscuit/cake/bread, candy/chocolate, fizzy drinks and fruit juice on 6-point response scales (twice or above per day, once per day, 3-4 times per week, once per week, once per month, or seldom or never). Reference to the calculation method is included in the text.

To elaborate:

Fresh fruits and sugar consumption were assessed as continuous variables. Sugar consumption was calculated based on overall frequency intake of four snacks and drinks (biscuits/cake/bread; candy/chocolate; fizzy drinks; fruit juice). For sugar intake frequency, each sugary item (snacks and drinks) was assigned a score as follows: twice or more a day (score of 2), once a day (score of 1), 3-4 times a week (3/7 days=0.43), once a week (1/7 days=0.14), once a month (1/30 days=0.03), and seldom or never (0). Weighted scores (the above values shown between brackets) have been chosen to match the lower frequency of consumption in each response category. Due to some missing cases of sugar intake frequency, mean substitution was used to impute missing cases. Weighted scores of four snacks and drinks were then summed to produce an overall score. The new variable was calculated from four snacks and drinks: sugar consumption= sum (biscuits/cake/bread, candy/chocolate, fizzy drinks, fruit juice), which ranged between 0 and 8. The score was used to indicate daily frequency of intake of sugars for each participant. Fresh fruit consumption was used as a separate variable using the same weights for frequency, and mean substitution was also used to impute missing cases.

  1. Line 94-96: Was this FFQ for a discrete time period, or generally? Recall bias is different depending on the way that the FFQ is administered. Was this a culturally appropriate FFQ? Was it validated, or how did foods end up on query?

Response to reviewer: We acknowledge the reviewer view on recall bias. The questions on diet used here are based on the Chinese National oral Health Survey, which is easier to use and understand by Chinese parents. The food frequency question was validated by the Chinese national oral health survey and used in previous studies.

  1. Line 128: Please include a diagram outlining participants included and excluded, with drop out reasons and demographics for children that might skew sample conclusions

Response to reviewer: A diagram outlining inclusion and drop out of this study is included in a previous paper, reference to the paper is inserted in the text. “Shen, Anqi, Eduardo Bernabé, and Wael Sabbah. The bidirectional relationship between weight, height and dental caries among preschool children in China. PloS one 14.4 (2019)”. Given that the diagram has been published before, we do not see the need to add it again here.

  1. Table 2: This could benefit from reorganization for visual appeal

Response to reviewer: Table 2 has been reorganized.

  1. I don’t know that the socioeconomic/intake index against anthropometric data is that important to show. This (anthropometric outcomes) could be assessed for covariance with dental caries scores through regression analysis, couldn’t they? What is the further impact of this data?

Response to reviewer: We appreciate the reviewer’s comment and agree about the importance of showing caries/anthropometric measures relationship. However, these relationships have already been assessed and published in a previous paper. “Shen, Anqi, Eduardo Bernabé, and Wael Sabbah. The bidirectional relationship between weight, height and dental caries among preschool children in China. PloS one 14.4 (2019)”. The objective of this manuscript though is to examine whether similar inequalities exist in both condition as stated in the introduction.

  1. Also, I do think that anthropometric outcomes need to be considered in modeling for the DMTF scores, as they may clearly have impact on baseline/ baseline to endline outcomes.

Response to reviewer: Thank you for the comment. As stated in the earlier point, this was not the objective of the current paper. The role of anthropometric measures in dental caries has been assessed in other papers. “Shen, Anqi, Eduardo Bernabé, and Wael Sabbah. The bidirectional relationship between weight, height and dental caries among preschool children in China. PloS one 14.4 (2019)”. The focus of this paper is only on socioeconomic inequality in caries and growth increment.

  1. Can authors consider doing a time modeling adjustment for DMTF outcomes and covariates?

Response to reviewer: In this study, we used multilevel analysis, which is considered appropriate for analysis of panel data. The multilevel model already accounts for time, two assessments of caries, two assessments of anthropometric measures. This is considered a more advanced analysis for longitudinal data than regression models, which account for baseline assessment.

Multilevel analysis is a methodology for the analysis of data with complex patterns of variability, with a focus on nested sources of such variability, such as longitudinal measurements of subjects. Longitudinal data are clustered since multiple observations over time are nested within units, typically subjects. Such clustered designs often provide rich information on processes operating at different levels. Multilevel modeling provides better estimates to address more complex questions.

  1. I think the analysis could use work overall: authors may benefit from looking at baseline outcomes, then endline outcomes adjusted for baseline, including anthropometrics, then change in baseline to endline outcomes (sort of like how researchers are looking at growth velocity now)

Response to reviewer: We would like to thank the reviewer for this point. As we elaborated in the response to the previous comment, multilevel analysis accounts for baseline, follow-up and time. It is also considered more appropriate for analysis of longitudinal data than regression analysis that use end point outcome, and adjust for baseline assessment and time.

Reviewer 2 Report

General

The aim of study is interesting and the strategy of investigation seems to be adequate. However, I think this article needs to a major revise of discussion and maybe conclusion. Authors insisted on that ‘socio-’, ‘economic’ factors are predictors for dental caries emergency and growth among preschool children. As results, the number of dental caries correlates to mother’s education, on the other hand, family income correlates to children’s growth. In other words, there is a significant correlation between (only) ‘socio-‘ factor and dental caries, and there is a correlation between ‘economic’ factor and growth. These two correlations seem to be mutually independent, and authors’ statements misleads readers to that there is correlation between economic status and dental caries. I do not think that the relation between family income and growth is main theme of this report because it seems to be obvious. Authors should refer and discuss about some literatures concern to the relation between parents’ education and dental caries (please see below).

Major concerns

  1. Line 89-91; Does ‘income’ mean monthly income? How much the mean income in this area is?
  2. The statement in Line 132-139 and Table1 are duplicate.
  3. Table2; Although the item of ‘10,000 or above’ correlated significantly to dmft in Model1, there is no discussion in the main text.
  4. Line 180-181; There was no correlation between income and dental caries. The state should be changed.
  5. Line 190-194; Authors should refer to and discuss about previous paper which showed correlations between dental caries and parents’ income or education, such as,

Cianetti et al, Dental caries, parents educational level, family income and dental service attendance among children in Italy. Eur J Paediatr Dent, 18:15–18, 2017.

Hooley et al, Parental influence and the development of dental caries in children aged 0–6 years: A systematic review of the literature, Journal of Dentistry, 40(11), 873-885, 2012.

  1. Line 222-227; Do you want to discuss about developing countries, or about China in this paragraph?
  2. Line 228-233; Please state the claim of this paragraph clearly.

Minor concerns

Line 115-126; Please state the name of statistical software.

Line 121; ‘z-score’ should be after ‘height-for-age’.

Line 144; CI should be spelled out.

Author Response

Response to reviewer’s comments:

We would like to thank the reviewer for the extensive effort in reviewing this paper. We have now addressed all the new reviewer’s comments and prepared a point by point response. Please find our response to your comments.

Major concerns:

  1. Line 89-91; Does ‘income’ mean monthly income? How much the mean income in this area is?

Response to reviewer: Income is monthly income. Income was based on the Chinese national oral health survey (4th Oral health survey). In 2019, the mean monthly income was 4,093 yuan in Liaoning Province.

  1. The statement in Line 132-139 and Table1 are duplicate.

Response to reviewer: We would like to thank you for raising this point. We have deleted some of the text and left some to highlight important findings.

  1. Table2; Although the item of ‘10,000 or above’ correlated significantly to dmft in Model1, there is no discussion in the main text.

Response to reviewer: This sentence was added “Only the highest income group (10,000 or above) was negatively and significantly associated with changes in dmft index (-0.90, 95% CI: -1.77, -0.02) in the semi-adjusted model. However, the association was insignificant in the fully adjusted model”.

  1. Line 180-181: There was no correlation between income and dental caries. The state should be changed.

Response to reviewer: We have changed this.

  1. Line 190-194; Authors should refer to and discuss about previous paper which showed correlations between dental caries and parents’ income or education, such as,

Cianetti et al, Dental caries, parents educational level, family income and dental service attendance among children in Italy. Eur J Paediatr Dent, 18:15–18, 2017. Hooley et al, Parental influence and the development of dental caries in children aged 0–6 years: A systematic review of the literature, Journal of Dentistry, 40(11), 873-885, 2012.

Response to reviewer: These two references have been added and discussed. The sentence was added “Education was considered as the most important indicator of socioeconomic inequalities in children’s caries, a finding consistent with this study”.

  1. Line 222-227; Do you want to discuss about developing countries, or about China in this paragraph?

Response to reviewer: We are discussing China, but illustrating that the situation there is different than that in developed countries. We changed the sentence to “Unlike in developed countries……”.

  1. Line 228-233; Please state the claim of this paragraph clearly.

Response to reviewer: We have revised this sentence to make it clearer.

Minor concerns:

Line 115-126; Please state the name of statistical software.

Response to reviewer: STATA was used to analyse the data.

Line 121; ‘z-score’ should be after ‘height-for-age’.

Response to reviewer: height-for-age z-score was added.

Line 144; CI should be spelled out.

Response to reviewer: CI (confidence interval) was added.

Reviewer 3 Report

Interesting study with important research area.

Some statistical consideration: As the study used Cohort data, the Statistical analysis suitable for longitudinal study was expected rather than Linear Regression Model. Moreover, time was not adjusted. It was not clear if it was Multi level or Multi variate Linear Regression. 

Secondly,it was not clear whether any dental caries was treated.

Thirdly, though it would be beyond the scope of the current research question, but it would be interesting to see the association of caries and height and weight in this population longitudinally. 

In current research question it feels as two seperate research question.

Author Response

Response to reviewer’s comments:

We would like to thank the reviewer for the extensive effort in reviewing this paper. We have now addressed all the new reviewer’s comments and prepared a point by point response. Please find our response to your comments.

1.Some statistical consideration: As the study used Cohort data, the Statistical analysis suitable for longitudinal study was expected rather than Linear Regression Model. Moreover, time was not adjusted. It was not clear if it was Multi level or Multi variate Linear Regression. 

Response to reviewer: Please see response to reviewer 1

Multilevel analysis was used in this study, which is a better for analysing longitudinal data. It was mentioned in page 3, Line 116, “Multilevel analysis was used to assess the association between baseline socioeconomic position and dental caries, weight and height”.

Multilevel analysis is a methodology for the analysis of data with complex patterns of variability, with a focus on nested sources of such variability, such as longitudinal measurements of subjects. Longitudinal data are clustered since multiple observations over time are nested within units, typically subjects. Such clustered designs often provide rich information on processes operating at different levels. Multilevel modelling provides better estimates to address more complex questions.

2.Secondly, it was not clear whether any dental caries was treated.

Response to reviewer: The use multilevel model accounts for change in dmft overtime as it uses two assessments of dmft. Having said that, dmft will always increase whether dental caries was treated or not (i.e., in dmft, if caries is treated, it will still be counted as filled or extracted. Still counted in dmft score).

3.Thirdly, though it would be beyond the scope of the current research question, but it would be interesting to see the association of caries and height and weight in this population longitudinally. 

Response to reviewer: The relationship between anthropometric measures and dental caries was assessed in the previous paper. “Shen, Anqi, Eduardo Bernabé, and Wael Sabbah. The bidirectional relationship between weight, height and dental caries among preschool children in China. PloS one 14.4 (2019)”. However, this paper only explored the socioeconomic inequality in caries and growth increment.

4.In current research question it feels as two seperate research question.

Response to reviewer: Yes, they are two separate questions, but the point we are trying to make is the similarity in inequality in indicators of oral and general health, in this instance, dental caries and children’s growth.

Reviewer 4 Report

Thank you for the opportunity to review this manuscript. The authors presented a longitudinal study of assessing socioeconomic inequalities in increment of dental caries and growth among preschool Chinese children.  

MS is generally well written, easy to read and it is understandable for the reader, however, some minor corrections/additions could be made.

As this is the first and only draft of this review, I apologize in advance for typographical and grammatical errors.

In section Materials and methods

Line 111, sentence about WHO Growth standard 2006 and 2007 seems incomplete finishing with “to”

Statistical analysis

I suggest to state statistical software used for calculations, and what p-value was considered statistical significant

Author Response

Response to reviewer’s comments:

We would like to thank the reviewer for the extensive effort in reviewing this paper. We have now addressed all the new reviewer’s comments and prepared a point by point response. Please find our response to your comments.

  1. In section Materials and methods: Line 111, sentence about WHO Growth standard 2006 and 2007 seems incomplete finishing with “to”

Response to reviewer: The sentence is “The WHO Growth Standards 2006 was utilized to convert weight and height measurements to z-scores weight-for-age and height-for-age among children aged 0-5 years old and WHO Growth Standards 2007 was used for children older than 5 years (18, 19). ”

2.Statistical analysis: I suggest to state statistical software used for calculations, and what p-value was considered statistical significant

Response to reviewer: STATA was used to do the analysis. P<0.05 was considered as statistical significant.

Round 2

Reviewer 2 Report

The revised manuscript was mostly improved. Please reconsider about the below which I wrote in the previous comment. I think the current title and abstract mislead conclusion to readers .

As results, the number of dental caries correlates to mother’s education, on the other hand, family income correlates to children’s growth. In other words, there is a significant correlation between (only) ‘socio-‘ factor and dental caries, and there is a correlation between ‘economic’ factor and growth. These two correlations seem to be mutually independent, and authors’ statements misleads readers to that there is correlation between economic status and dental caries. 

Author Response

We would like to thank the editor and the reviewer.

We have added the following statement to the results in the abstract: Family income was not significantly associated with dmft in the fully adjusted model.

The reviewer said the title and abstract are misleading and suggested that dmft is associated with only ‘socio’ factors and growth associated with ‘economic’ factor.

In the revised abstract now we make it very clear that dmft was not associated with income in the fully adjusted model.

If the reviewer is referring to the inclusion of education under the term socioeconomic factor, as it appears in the title, education has always been an indicator of socioeconomic factor as it is usually related to income. We are listing a number of references that clearly state that education is an indicator of socioeconomic factor, including reports from WHO and American Psychological Association (through the link).

Duncan GJ, Daly MC, McDonough P, Williams DR. Optimal indicators of socioeconomic status for health research. Am J Public Health. 2002;92:1151–1157.

Darin-Mattsson A., Fors S., Kåreholt I. Different indicators of socioeconomic status and their relative importance as determinants of health in old age. Int. J. Equity Health. 2017;16:173. doi: 10.1186/s12939-017-0670-3

Assessing the distribution of health risks by socioeconomic position at national and local levels. Environmental Burden of Disease Series, No. 10 WHO 2014 Available form: https://www.who.int/quantifying_ehimpacts/publications/ebd10.pdf

American Psychological Association. Measuring Socioeconomic Status and Subjective Social Status. Available from: https://www.apa.org/pi/ses/resources/class/measuring-status